# A model of anaerobic tissue perfusion during trauma—Lactate trajectory curvature can determine recovery

Austin Baird[1]*, Steven A. White[2], Erika K. Bisgaard[3], Rachel K. Wentz[4], Edward M. Sims[4], David Hananel[1]

**1** University of Washington Department of Surgery, Division of Healthcare Simulation Sciences, Seattle, Washington, United States of America, **2** Applied Research Associates, Southeast Division, Raleigh, North Carolina, United States of America, **3** University of Washington Department of Surgery, Division of Trauma, Burn, and Critical Care Surgery, Seattle, Washington, United States of America, **4** Vcom3D, Inc., Orlando, Florida, United States of America

\* abaird1@uw.edu

## Abstract

Hypovolemic shock and hemorrhage continue to shape healthcare delivery and incur a heavy burden on public health and wellbeing. Many healthcare organizations have specific transfusion protocols that are referenced when a patient meets certain physiological criteria. These protocols are shaped around best practices and nuanced understanding of the physiology of the patient. Despite their importance, these protocols are extremely difficult to change or challenge and, to date, there does not exist a sufficient mathematical model that may be employed towards investigating simulated patient hemorrhage. We show that once constructed, we can consider the health of the patient in phase-space rather than traditional time-series representations, quantifying the recovery of the patient due to the geometry of the trajectories. We construct a whole-body physiology model of hemorrhagic shock and trauma that encompasses multiple organ systems, non-linear feedback mechanisms and patient variability. We validate this model by constructing a major hemorrhage scenario, that includes transit time and associated mass transfusion resuscitation of the patient. We then use this model to create phase-plane diagrams of patient trajectories as a function of lactate blood pH and volumes, among other relevant physiological metrics. Exploring these patient trajectories amongst a varied patient population yields a series of high curvature points associated with the transition from deterioration to recovery of the patient. We then construct a convex hull, covering the high curvature regions of a diverse simulated patient population. These hulls, constructed by simulated numerous patients, exposed to three distinct hemorrhagic severities, are then used to validate the model by comparing experimental serum lactate levels as a function of blood volume to these regions in phase-space. In conclusion, we show that the model

**Data availability statement:** All data and code required to re-create the studies described in this manuscript can be found here: https://github.com/ajbaird/PaperMaterials/tree/main/hemorrhagepH.

**Funding:** Adaptation and validation of the model for simulating trauma patient cases was funded in part by a Defense Health Agency (contract #W81XWH22C0137 DHA E.S. and R.W.) Small Business Innovation Research awarded to Vcom3D, Inc., and monitored by the US Army Combat Capabilities Development Command Soldier Center (CCDC SC) SFC Paul Ray Smith Simulation and Training Technology Center (STTC). The funders had no role in study design, data collection and analysis, decision to publish, or preparation of the manuscript.

**Competing interests:** The authors have declared that no competing interests exist.

is highly accurate and accounts for distinct lactate trajectories amongst the simulated patient cohorts. Finally, we fit a logistic curve to the resulting data for a quick patient severity tool and denote the standard error of parameterization based on the delta method.

## Author summary

Hemorrhage is one of the leading causes of death and morbidity across the globe. The consequence of massive hemorrhage is significant changes in the human physiology, resulting in numerous feedback mechanisms working towards keeping various organs working properly, despite the loss of blood volume. Due to the burden on the body, rapid resuscitation of the patient is paramount to survival with numerous studies showing that cadence, volume, and timing of infusions leading to improved patient recovery. We aim to construct a computational capable of simulating the complexities of this event with the ability to also capture the recovery phase due to a specific transfusion protocol. This platform, we show, can be used to capture patient degradation in the face of this loss of volume but also recovery due to surgical intervention and infusion of blood products. We hope that this model can be used to create an in-silico patient model that can be used as a first step in designing clinical trials.

## 1 Introduction

Hemorrhage, characterized by the loss of blood from the circulatory system, is a critical physiological event often encountered in trauma, surgery, and various pathological conditions. Its consequences extend far beyond mere blood loss, triggering a cascade of systemic responses that profoundly impact cardiovascular function, tissue perfusion, and metabolic homeostasis [1–5]. Concomitantly, hemorrhage-induced acid-base disturbances play a pivotal role in shaping the physiological response to blood loss, influencing cellular metabolism, oxygen delivery, and overall systemic equilibrium [6–8]. Understanding the intricate interplay between hemorrhage and acid-base disturbances is crucial for elucidating the mechanisms underlying physiological compensation and decompensation in response to blood loss. Moreover, accurate modeling of these phenomena holds significant clinical relevance, aiding in the prediction of patient outcomes, optimization of resuscitative strategies, and refinement of therapeutic interventions [9–12].

Despite advancements in our understanding of hemorrhagic shock and acid-base disturbances, significant gaps persist in both clinical practice and modeling approaches. Firstly, current clinical guidelines for managing hemorrhagic shock often lack specificity regarding the optimal resuscitative strategies to mitigate associated acid-base imbalances [13–15]. While fluid resuscitation remains a cornerstone intervention, the choice of fluid type, volume, and timing remains contentious, particularly

in the context of concurrent acid-base disturbances. Various recommended treatments may cause additional harm when administered [16,17]. For example, transfusion protocols have evolved over the past century to recommend permissive hypotension, and limited crystalloid administration [18]. Moreover, the individualized nature of patient responses to resuscitative efforts necessitates a more nuanced approach that integrates dynamic physiological parameters, such as tissue perfusion, oxygen delivery, and metabolic status.

Existing work investigates lactate kinetics in different contexts, such as lactate metabolism in animals with a focus on experimental support [19], exploring lactate kinetics under trauma-relevant inflammatory conditions using a systems biology approach [20], and lactate shuttling and metabolic flux relevant to both exercise and injury physiology [21,22]. These studies provide important frameworks for understanding lactate production, clearance, and redistribution. Our model extends this literature by integrating these dynamics into a trauma-specific physiological context with real-time adaptability for personalized monitoring and coupling lactate dynamics directly to the cardiopulmonary system and blood pH.

We note that, existing models often oversimplify the complex interplay between hemorrhage and acid-base disturbances, overlooking key determinants or feedback mechanisms, often focusing on one singular mechanism as opposed to integration and connection of multiple processes [23]. Many models predominantly focus on cardiovascular hemostasis, not stressing systems into circulatory shock [22,24,25]. This fails to capture the complex physiological nature of hemorrhagic shock and its impact on acid-base balance and peripheral systems. Consequently, there is a pressing need for integrative modeling approaches that account for the dynamic interactions between cardiovascular, respiratory, nervous, and metabolic systems during hemorrhage-induced acid-base perturbations. The aim of models of this type is facilitation of the development of more effective resuscitative strategies and analysis of synthetically generated data that may be tailored to individual patient profiles.

In this paper, we present an integrative modeling framework aimed at elucidating the complex interactions between hemorrhage and acid-base disturbances in physiology. Leveraging computational techniques and mathematical simulations, our approach offers a comprehensive examination of the dynamic physiological processes that unfold during hemorrhagic events, encompassing alterations in cardiovascular dynamics, fluid shifts, tissue oxygenation, and acid-base balance. We formulate this model as a distributed lumped-parameter 0D physics-based circulatory system, coupled to compartmental mechanistic models of specific organs and physiological functions. Coupling these two modeling frameworks effectively constructs a whole-body, multidomain model, while omitting 1D spatial component from the fluid dynamics. Central to our modeling paradigm is the incorporation of established physiological principles, and principles of oxygen transport and consumption. By integrating these concepts within a cohesive computational framework, we aim to provide a nuanced understanding of the multifaceted responses elicited by hemorrhage and their impact on systemic acid-base equilibrium.

For this research effort, we add to the existing BioGears physiology model in two distinct ways. We add a model of acid-base that includes adjustments due to hypovolemia by reacting to nonlinear lactate perturbations. We leverage the existing heart driver, nervous system, and circulatory system to construct an investigation into how these existing models respond to severe hypovolemia and associated acid-base disturbances.

We use this model to generate synthetic data and analyze this data in the form of patient trajectories. These trajectories may inform recovery characteristics and are not able to be properly studied in experimental patient data due to the lag time involved in laboratory measurements of patient blood. We show that for worsening hemorrhage there is a distinct shape to the maximum curvature point in the patient trajectory. This curvature is studied over a diverse patient population, and we report statistics on its variance for this given population. We show that for these statistics, patient recovery can be defined by a curve fit to the maximum curvature for a given patient hemorrhage, creating a potentially useful clinical measurement tool, as lactate measurements, coupled to a more available measurement such as blood pressure, can be analyzed against this curve to determine patient state.

## 2 Materials and methods

We construct a model of the cardiopulmonary system coupled to oxygen diffusion, blood-gas binding and transport, anaerobic tissue perfusion, and nervous system autoregulation in the BioGears physiology engine [26] as this platform provides coupling to various other systems in the body to determine global physiological patient response. In addition, BioGears has a robust application programming interface (API) that allows for programming diverse patient trauma scenarios and extraction of data broadly (heart rate) or at a more refined level (lactate concentration in the renal tubules). The BioGears physiology engine has been successfully used in modeling of sepsis [27], burn [28], hemorrhage [29], and used as the patient physiology in many healthcare simulation studies [30,31].

### 2.1 Cardiopulmonary system—Circulation and respiration

Following the work done in whole-body physiological models that include circulation [32–35], we model the circulation by constructing an electrical circuit analog that characterizes the fluid dynamics, Fig 1. Flow rates and pressures are approximated by current and voltage and computed each time step. Using this approximation, we compute the state of the circulation leveraging Kirchhoff's voltage and current laws at each node on the graph. We begin by denoting ground as our reference node the label nodes in the system and initialize certain node pressures as a function of experimental data [36,37] or are optimized for desired patient dynamics. Once nodes are labeled and initialized we perform modified nodal analysis [38] to populate a matrix that contains the system dynamics. For each node we use Kirchhoff's current and voltage laws to solve for unknown quantities. If Voltage is defined across a resistor, we use Ohm's law to define a relationship between current and voltage. Using this denotation, we construct the following matrix equation:

$$\begin{pmatrix} G & B \\ C & D \end{pmatrix} \begin{pmatrix} V_n \\ I_m \end{pmatrix} = \begin{pmatrix} I_s \\ V_s \end{pmatrix}$$

Here, the G matrix is the nodal admittance matrix (conductance matrix for resistors). B is a matrix representing the connection of voltage sources to nodes. C is the transpose of B. D is a diagonal matrix representing the internal resistances of voltage sources. $V_n$ is the unknown voltages and $I_m$ denotes the unknown currents. $I_s$ and $V_s$ denote the current and

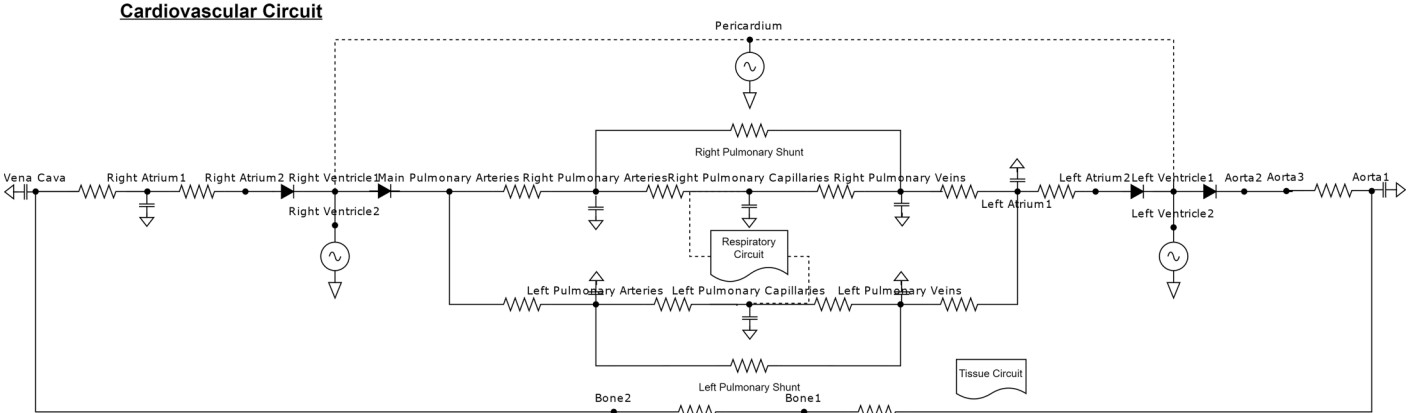

**Fig 1. Cardiovascular circuit structure for the BioGears circulatory system.** Respiratory, cerebral, tissue, and renal circuits are omitted for brevity. Left and right ventricle drive the system through elastance changes. Pulmonary arteries oversee oxygen diffusion between the respiratory (gas) circuit into the cardiovascular (liquid) circuit.

voltage sources in the system. For our implementation, we note that $I_s$ is zero. To handle the compliances in the system we approximate using backward Euler step to compute the current for a given capacitor C:

$$I_C(t + \Delta t) = C \frac{V_C(t + \Delta t) - V_C(t)}{\Delta t}$$

Thus, for each capacitor connecting nodes i and j, we have:

$$I_i(t + \Delta t) = C \frac{V_i(t + \Delta t) - V_j(t + \Delta t)}{\Delta t}$$

Now defining $G_C = \frac{C}{\Delta t}$, we can derive an update scheme for the G and Z matrices to be:

$$G_{ij} = G_{ij} - G_C$$

$$Z_i = Z_i + G_C V_j(t)$$

$$Z_j = Z_j - G_C V_i(t)$$

We note that along the diagonal, we sum the capacitance contribution in the G matrix. Once the matrix is constructed, we use LU factorization to solve the sparse system.

To drive the circulation, building on prior work [39], we construct a driver function that relates elastance to cardiac cycle time:

$$E_v(t) = (E_{max,v} - E_{min,v}) \left( \frac{f(t)}{f_{max}} \right) + E_{min,v}$$

Here $E_v$ denotes the elastance for a given ventricle with associated max and min values and f is a nonlinear relationship between current simulation time and period of contraction for a given cardiac cycle:

$$f(t) = \left( \frac{\left( \frac{t}{\alpha_1 T} \right)^{n_1}}{1 + \left( \frac{t}{\alpha_1 T} \right)^{n_1}} \right) \left( \frac{1}{1 + \left( \frac{t}{\alpha_2 T} \right)^{n_2}} \right)$$

Here T is our cardiac cycle length and alpha and n denotes shape parameters that are fit to a pressure volume curve.

The gas transport equations for the respiratory circuit are identical to the cardiovascular system, with one exception, the driver. We derive our driver function to relate the respiratory muscle pressure (negative pressure) to the respiratory cycle time:

$$P_{musc} \begin{cases} -\frac{P_{max}}{IE} t^2 + -\frac{P_{max} T}{IE} t, & 0 < t < I \\ \frac{P_{max}}{1 - e^{\left( \frac{-E}{\tau} \right)}} \left( e^{\left( \frac{t - I}{\tau} \right)} - e^{\left( \frac{-E}{\tau} \right)} \right), & I < t < T \end{cases}$$

Here I, E, and T are the inspiratory, expiratory, and total respiration times, respectively. The value $\tau$ is a time constant for the expiration period and is estimated as E/ 5. The total breathing cycle time T is obtained from the inverse of the respiration rate determined by the chemoreceptor model (omitted from this paper for brevity). I and E are calculated using the inspiratory: expiratory ratio from the previous time step, which is modified by irregular physiology like asthma and COPD. The chemoreceptor model also updates the driver amplitude, $P_{max}$. The baseline value of $P_{max}$ for each virtual patient is

determined during engine initialization by modifying the amplitude at the requested patient respiration rate until a stable tidal volume is obtained. These drivers are both manipulated by the nervous system depending on the state of the patient. For our hemorrhage model, we develop the baroreceptors to buffer the pressure drops occurring due to blood loss.

## 2.2 Baroreceptor response

The baroreceptor mechanism provides rapid negative feedback control of arterial pressure. A drop in arterial pressure is sensed by the baroreceptors and leads to an increase in sympathetic activity and vagal (parasympathetic) withdrawal. These changes operate with the goal of maintaining arterial pressure at its healthy resting level. We distinguish between aortic, carotid, and low-pressure (cardiopulmonary) receptors. Aortic and carotid receptors are both sensitive to changes in systolic arterial pressure, but their relative locations in the body affect this response. Aortic baroreceptors respond to the transmural pressure between the aorta and the intrapleural space. The carotid baroreceptors, located a distance above the heart, are affected by pressure changes, generally increases, in the carotid artery. Low-pressure receptors are located near the venous return to the heart and are therefore sensitive to the central venous pressure and pleural pressure. We use a stress-strain relationship to calculate the signal generated at the aortic and carotid baroreceptors [40] and a first-order, low-pass filter to generate the low-pressure receptors signal [41]. Thus, we describe the strain exerted on aortic and carotid baroreceptors as:

$$\varepsilon_w = 1 - \sqrt{\frac{1 + exp\left(-q_w\left(P_{input} - s_w\right)\right)}{A + exp\left(-q_w\left(P_{input} - s_w\right)\right)}}$$

Here $\varepsilon_w$ is the wall strain, $P_{input}$ is the systolic pressure for the carotid baroreceptor and the difference between systolic and plural pressure for the aortic baroceptors, A is the maximum stressed to unstressed vessel cross-sectional area, $q_w$ is the steepness of response, and $s_w$ is the operating point of the baroreceptor response. The operating pressure is the systolic pressure after initialization of the patient. We note that the operating point may also be influenced by drugs, pain, or exercise although these adjustments were not considered for this manuscript.

We model adaptation [42,43] by adjusting the operating point as a function of systolic arterial pressure:

$$\frac{ds_w}{dt} = k_{adapt}\left(P_{sys} - s_w\right)$$

Here the value of $k_{adapt}$ is chosen to generate an approximately 16-hour half-life for the baroreceptors to adapt to a step change in systolic pressure. We further model the baroreceptor response as a spring dashpot system where we represent the strain induced by a stress in the system to be:

$$\frac{d\varepsilon_b}{dt} = \frac{-\varepsilon_b + k_b\varepsilon_w}{\tau_b}$$

Here the strain, $\varepsilon_b$ is processed in the central nervous system. To model the low-pressure baroreceptor signal generation, we introduce a low pass filter that samples the transmural pressure between the venous return (CVP) and the pleural space:

$$\frac{dP_{cp}}{dt} = \left(\frac{1}{\tau_{cp}}\right)\left(-P_{cp} + \left(CVP - P_{plueral}\right)\right)$$

And connect this model to the firing rate:

$$f_{cp} = \frac{f_{cp,max}}{1 + exp\left(\left(CVP_{base} - P_{plueral,base}\right) - P_{cp}\right)}$$

Here $f_{cp}$ is the firing rate processed by the CNS. $\tau_{cp}$ and $f_{cp,max}$ are the time constants for the response and the maximum firing rate, while $CVP_{base}$ and $P_{pleural,base}$ represent the normal central venous and pleural pressures. Following similar work [23,24,44,45], we then weight the signal produced in the afferent arm of the central nervous system to produce the final afferent signal. We omit model details of the peripheral chemoreceptors, and pulmonary stretch receptors as they are not relevant for the current study.

## 2.3 Hypoperfusion model

For a given tissue compartment, during circulation, the cells request a baseline level of oxygen to support ATP creation. During periods of hypoperfusion, the tissue reverts to anaerobic activity, generating lactate. This generation perturbs the ph value of the blood, resulting in changes in the biding activity of red blood cells [7,8,46]. We generate a model based upon prior work, that couples the strong ion difference in the blood, ph, and (as circulated by the cardiovascular model) bound and unbound o2 bicarbonate and co2:

$$SID = C_{Na}(t) + C_K(t) - (C_{Cl}(t) + C_{lactate}(t))$$

$$0 = SID - C_{HBCO3}(t) - C_A(t)(0.123pH - 0.631) - C_{PO4}(t)(0.309pH - 0.469)$$

$$0 = T_{CO2} - C_{CO2}(t) - C_{HBCO3}(t) - 4S_{CO2}C_{Hb}(t)$$

$$0 = T_{O2} - C_{O2}(t) - 4S_{O2}C_{Hb}(t)$$

$$pH = 6.1 + log\left(\frac{C_{HBCO3}(t)}{\alpha P_{CO2}(t)}\right)$$

We solve this system for its root at each time step for each liquid compartment in the cardiovascular system to determine oxygen binding during a hemorrhage event. To further perturb the anaerobic activity, we also scale lactate levels as a function of total blood volume in the patient:

$$L(V_{BV}, t) = \frac{15.5}{1 + 5.4exp\left(25\left(\frac{V_{BV}(t)}{V_{BV,baseline}} - 1\right)\right)} + 1$$

L is lactate mass, and V denotes blood volume. Parameter values for this expression were chosen to validate well with experimental data (see results section). Due to the major role the kidneys play in filtering lactate from the blood and adjusting pH balance in the blood [47], we introduce a relationship between mass transported into the renal tubules and blood pH:

$$M_L(pH(t), t) = \frac{10.0}{1 + 24.5exp(100(pH(t) - 1))} + 1$$

Here M denotes the mass of lactate in the renal capillaries, and pH denotes the pH in the blood. Parameters were qualitatively measured to capture recovery time post bleed. This value scales the amount of lactate filtered by the kidney's by moving mass into the tubules for clearance. At each time step we move lactate mass into the tubules:

$$\frac{dM_{sub}}{dt} = M_L C_{sub}\beta\gamma J(t)$$

Taken all together the BioGears physiology model is executed each time step in the following order: preprocess, process, and postprocess. The preprocess steps execute all system level models, beginning with the cardiovascular system, proceeding to the blood gas binding and pH calculations, and finishing with the nervous system models. These values then update resistances and compliances in the fluid circuit. Using these adjustments we then solve the fluid circuit during the process phase of execution. Finally, the postprocess phase pushes the new circuit values out to the distributed system, providing updates to pressures and flow rates, Fig 2. Time stepping of the model differential equations uses a global time step of 0.05s, which provides resolution of the heart rate frequencies seen during hypovolemia and ensure stability of each submodel. Future work will investigate variable time stepping with model, specific executions to speed up computation time.

## 2.4 Parameterization of the model

Parameterization of the lactate release model was dependent on the physiological impact of hemorrhage and associated hypoperfusion of the patient. Identifying parameters that could the produce lactate levels as measured in serum sampling in identified experimental manuscripts proceeded as an optimization process surrounding minimizing the residuals on the model of lactate release as a function of blood pH. Here we define the residuals between the maximum lactate value produced for a given hemorrhage and hypoperfusion amount (simulated) and associated experimental study. This effectively produced a set of parameters for the given $L(V_{BV}, t)$ equation. As there was large variance found in the experimental

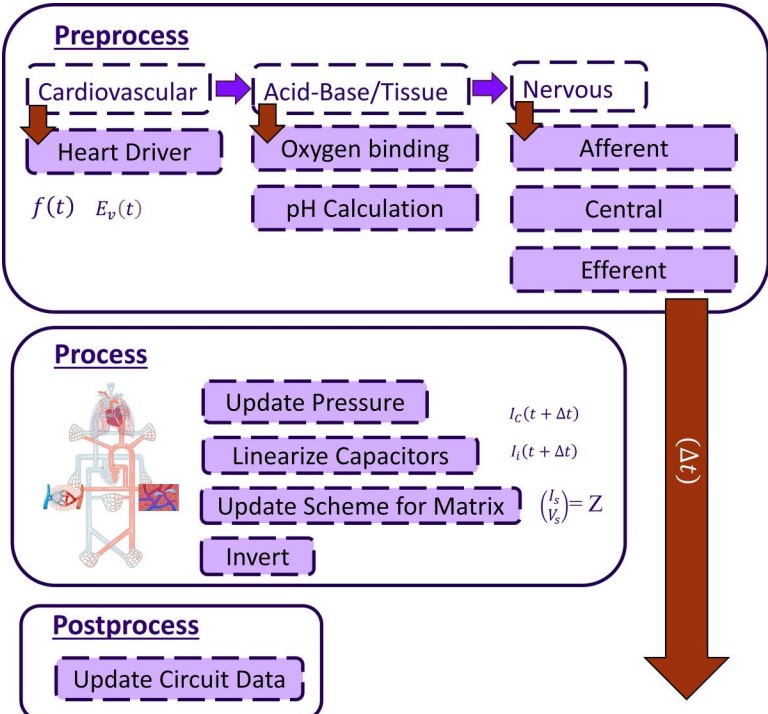

**Fig 2. Overview of the implemented model of physiology.** A time step in the model proceeds down in the diagram from preprocess to postprocess. Within preprocess there is an ordering of system execution that proceeds left to right. The heart driver updates elastance as a function of the heart cycle, then the oxygen binding is computed and associated pH calculations. Here the hypovolemia model implements lactate generation. Next the nervous system responds to hypovolemia by adjusting the baroreceptor response. Next the circuit is solved for pressures and flow rates and finally postprocess updates the circuit data values based upon the matrix inversion.

values for a given hemorrhage, we computed the minimized residual based upon the mean value computed for each of the 4 validation studies.

The models of the nervous system and computation of the pH balance, parameters were sampled from known prior models, with minimal qualitative adjustments made to fit within the context of the BioGears engine other existing models, Table 1. Future work will focus on identification of parameter sensitivity in the context of hemorrhagic shock, which will provide biological interpretability towards patient trajectories and the various subsystems that may overdetermine outcomes.

## 2.5 Calculating curvature of trajectories

We have now constructed a model of the hemodynamics of the patient, given stresses introduced by hypovolemia. We effectively connected the circulation to a regulatory model of the nervous system, able to buffer changes in arterial and venous pressures. Using this we may extract time series data from the simulation during a hemorrhage event. For a given trajectory that we will define as

$$\hat{r}(t) = (\hat{x}(t), \hat{y}(t))$$

Given as time series data where the x,y points consist of patient physiological data extracted from the physiology model. We assume x,y and only dependent upon t and may consist of things like: heart rate, respiration rate, lactate concentration in the blood, etc. For this trajectory, to extract maximum curvature value during trauma and subsequent recovery, we first smooth the data using a convolution for each array x,y over a discrete window:

$$(x * 1_m)[n] = \sum_{m=-M}^{M} x[n-m] 1_m[m]$$

Here 1 denotes a normalized ones vector over our window size, m, and x can be interchanged with any physiological variable we are extracting from the model. This operation effectively smoothing the data over the window size. This operation is critical as a curvature algorithm will converge to high frequency found on the trajectory path, not necessarily the global curvature maximum over the simulation. After smoothing we compute the curvature along the trajectory:

$$k = \frac{\left| \dot{x}(t)\ddot{y}(t) - \dot{y}(t)\ddot{x}(t) \right|}{(\dot{x}^2(t) + \dot{y}^2(t))^{\frac{3}{2}}}$$

From this expression we can extract the maximum curvature over a given trajectory to determine the inflection point in which the patient begins their recovery. We aim to analyze this point over the x,y domain to segment the plane into disease state.

**Table 1. Parameter values for the nervous system models as they respond to pressure stimulus. Signals are sent to update cardiovascular system. Parameter values were taken from prominent prior results with minimal qualitative adjustments.**

| Parameter | Value | Unit | Description | Citation |
|---|---|---|---|---|
| $-q_w$ | 0.04 | 1/ mmHg | Steepness of the arterial wall strain baroreceptor response | [48] |
| $A$ | 0.5 | Dimensionless | Maximum to unstressed cross-sectional area | [48,49] |
| $\tau_b$ | 0.9 | S | Time constant | [50,51] |
| $k_b$ | 0.1 | Dimensionless | Gain | [48] |
| $\tau_{cp}$ | 6.0 | S | Time constant | [41] |
| $f_{cp,max}$ | 20 | S | Time constant | [48] |

## 2.6 C++ implementation

The BioGears Physiology Engine is an Open-Source C++ based numerical physiology simulation provided through the public repository on GitHub. The core software is licensed under the APACHE 2.0 permissive license and can be included in any project provided public attribution is presented.

The libraries provided by the BioGears project are divided into three primary components. The Common Data Model (CDM), Synthetic Environment (SE) interface layer, and the BioGears Engine human physiology implementation layer. The CDM and SE layers of the architecture allow the development and communication of physiological concepts across multiple computing architectures and facilitate snapshotting virtualized patient states for repeated use and review, Fig 3. Additional details on the purpose of this division has been discussed in prior work [26,27], for this work we will primary be discussing the BioGears Engine implementation as the work described is primarily encompassed at this architectural level.

The BioGears Physiology Engine is an object-oriented electrical circuit analog for characterizing the multi-faceted fluid dynamics of a human physiology model. BioGears makes heavy use of C++ template meta programing for its underlying representations of both circuit analogs and fluid compartments. The template network describes the full circuit model of the currently simulated human physiology overlayed with physiology compartment divisions which track and respond to fluid and substance transfer within the system. At the end of each calculation cycle the interfaces provided by the SE sub-system allow integrators to probe the complete status of the virtual patient at multiple levels of resolution allowing for the development of immersive user facing medical training, simulation, and exploratory solutions.

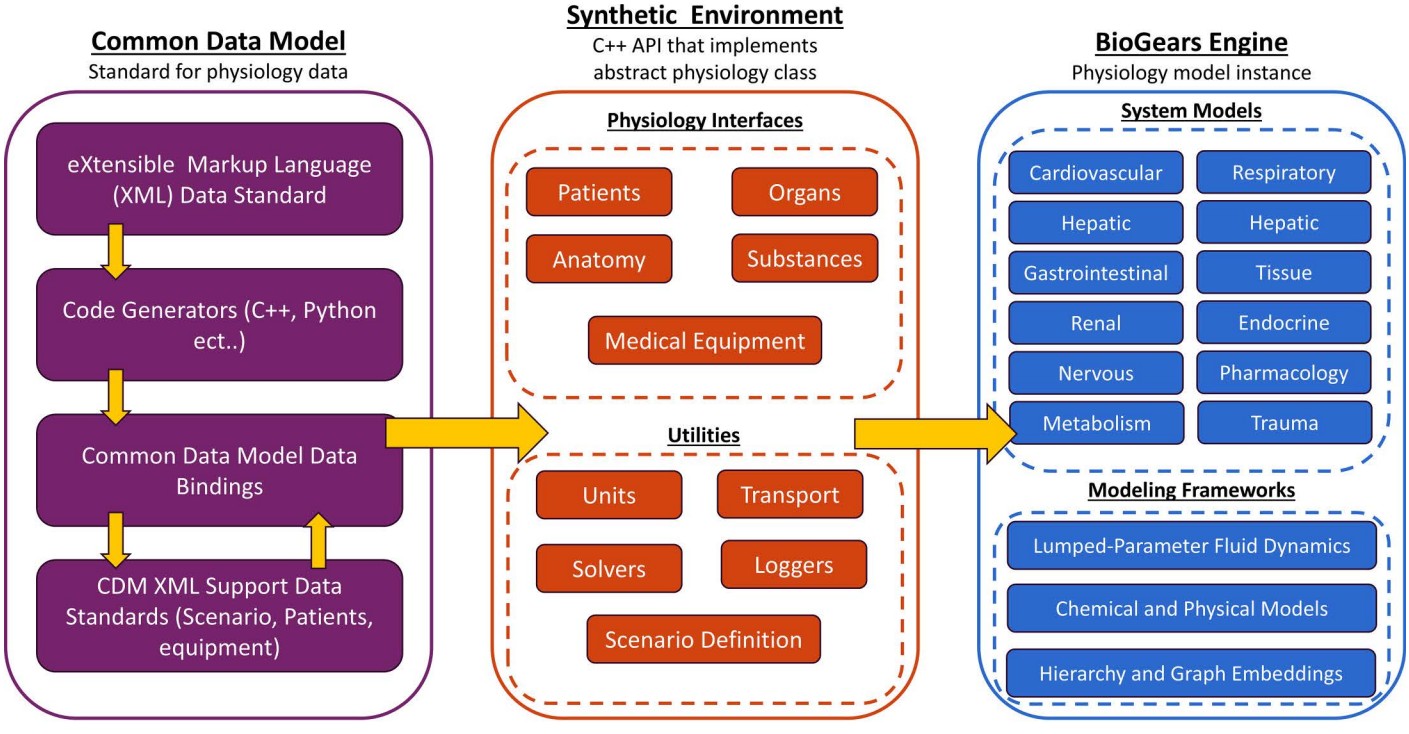

**Fig 3. Overview of the C++ structure of the BioGears physiology engine.** Common data model handles data input and output as well as standard-ization of inputs such as patient definition. The synthetic environment wraps the models to generate a complete application programming interface. The engine implements a model of the physiology for a given patient.

## 3 Results

### 3.1 Physiology of hemorrhage

We begin by configuring the patient with a series of increasingly severe hemorrhages and investigate the overall cardiovascular patient physiology as a function of this severity. In the constructed scenario, we initialize the hemorrhage, and then proceed to let them bleed for 10 minutes which we assume to approximate a standard emergency transit time. Once the patient has been successfully transferred to a healthcare institution, we begin a transfusion protocol, in line with hemorrhage severity [10,14,18,52]. After an additional 10 minutes of fluid resuscitation, we assume the patient has had their bleeding stopped via a surgical intervention and continue resuscitation until the shock index, here defined to be the ratio between heart rate and systolic pressure, is below 1 [53–55]. We note qualitative validation of the behavior of our model through the generalized increase of heart rate and decreases in blood pressure pH and blood volume, Fig 4. We note that the recovery phase in physiological response to hemorrhage is much longer, as the baroreceptors continue to remain diminished until blood volume and pressure recover fully during the scenario.

Although time series data is generally presented with a dependent (such as pH) and independent variable (such as time), we aim to discover geometric coverage that may predict patient state beyond time dependance. Indeed, the idea of geometric dependance on biological function has been used in prior work where state space is used to quantify relationships between experimental and model data [56–59] and is argued to be essential in analysis of biological systems

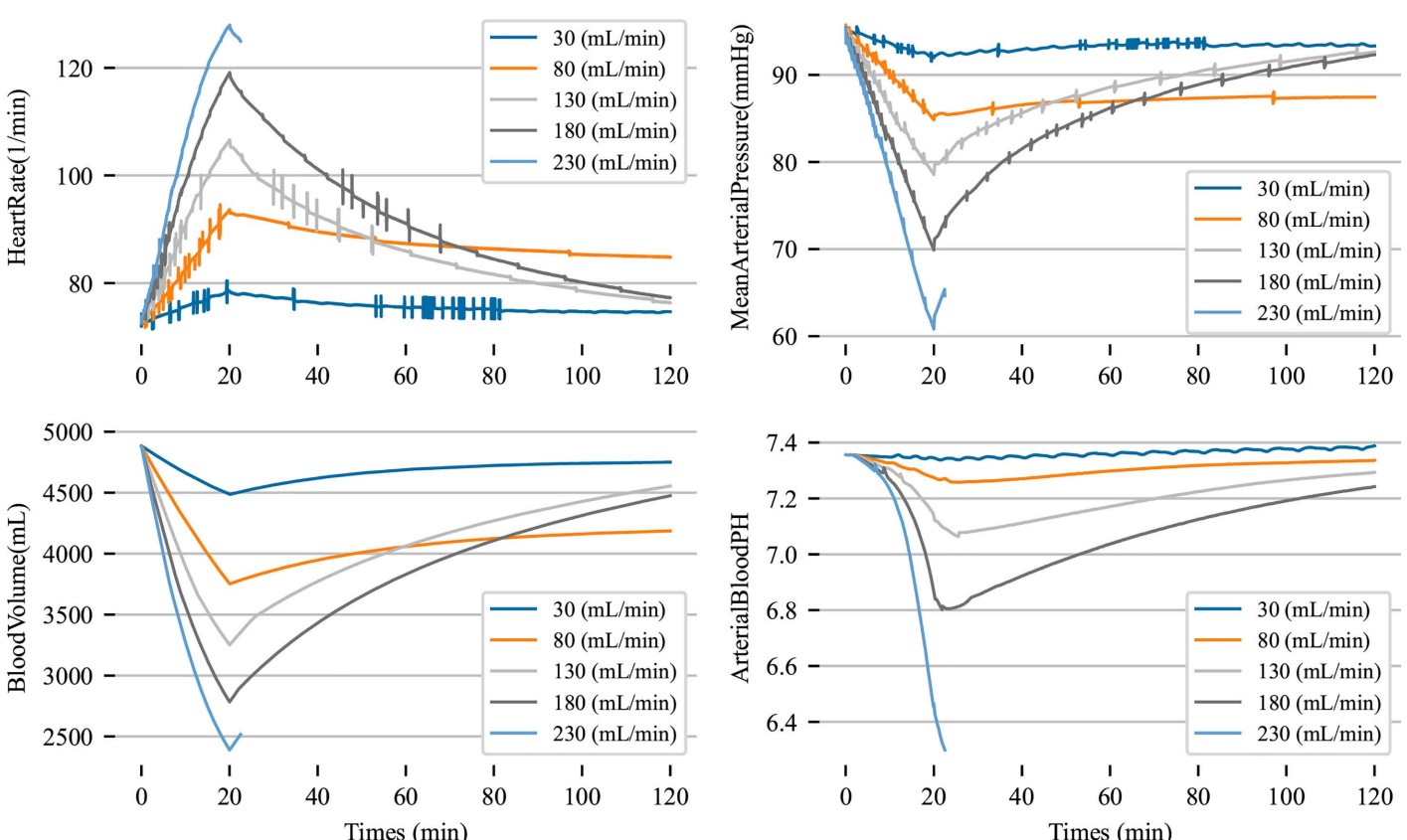

**Fig 4. Cardiovascular metrics reported for varying levels of initial hemorrhage.** Patient time series data for the nervous system response and lactate molarity due to hemorrhage for a single patient over multiple initial bleed rates.

. Time often obscures the relationships between physiological variables and doesn't allow for proper analysis of such relationships. To do this we consider the trajectory of patient vitals as a coupled data set $\hat{y} = (y_1(t), y_2(t))$. Here 1 and 2 may denote two physiological variables collected from the model simulation, such as lactate concentration and pH or blood volume and mean arterial pressure. We aim to compute the moment of greatest curvature along these trajectories to fit a nonlinear model to these points to predict patient state.

## 3.2 Patient trajectories

We begin by identifying three trajectories of interest, with lactate concentrations being our primary target of investigation. These trajectories include respiration rate, pH, and lactate concentration, with all three coupled to the blood volume as $y_2(t)$ for a given y hat trajectory, Fig 5. Due to oscillations seen in the data we smooth out the local curvature points and smooth each trajectory. We then can collect the maximum curvature over the entire trajectory of the smoothed curve. We note different non-linear behavior over the set of curves presented here. Because the recovery phase of the model begins after a set amount of time, the blood volume is not the same amongst models. Using this presentation we can see a distinct nonlinear trajectory, particularly for the lactate concentration and corresponding pH. Here we compute the pH of the patient using the strong ion difference formula: $pH(t) = 6.1 + log\left(\frac{SID - [A^-]}{0.03 pCO2}\right)$. We note that computed pH is a function of the lactate which is changing considerably throughout the simulation. Other ions in the blood are staying relatively constant.

Noting this nonlinearity, we want to investigate the variability of the curvature amongst a collection of patients then proceed to fitting the curvature to give a determination of predicted patient state. We collect a random sampling of patient variables, allowed by the BioGears physiology engine, that are within "normal" physiology for an adult. These variables include resting heart rate, baseline arterial pressure, sex, resting respiration rate, and heart rate, Table 2. These patient

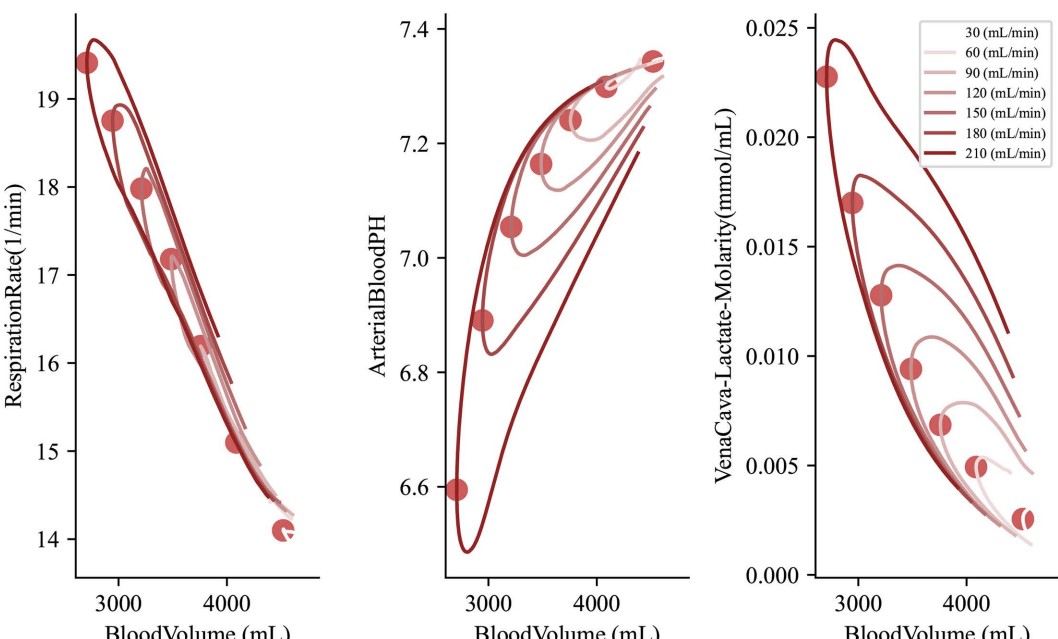

**Fig 5. Patient trajectories over multiple hemorrhage events.** Figures are shown after smoothing. Dots denote the computed max curvature for the given trajectory. Blood volume was chosen as a parameter for each physiology value to compare results. A non-linear trend to each set of maximum curvature points is denoted amongst the plots.

**Table 2. Descriptive statistics for the simulated patient population.** The targeted cohort was constructed to be a healthy middle age population with no comorbidities. Tight variance was arbitrarily chosen for this investigation but cohort development, to understand model sensitivity, is something that is supported by the BioGears engine.

| Parameter | Description | Mean | Variance |
|---|---|---|---|
| Sex | Male/Female | 50-50 split | N/A |
| Age | years | 36.9 | 3.1 |
| Weight | Kg | 69.2 | 13.5 |
| Diastolic Pressure | mmHg | 72.8 | 2.1 |
| Systolic Pressure | mmHg | 114.3 | 1.7 |
| Resting Heart Rate | Beats/min | 71.3 | 2.5 |
| Resting Respiration Rate | Breathe/min | 15.5 | 2.1 |

variables are used as an input to the physiology engine and augment various cardiopulmonary volumes depending on their values. We note that the variance of these values, for this constructed virtual patient cohort, is quite narrow and only reflects a very specific healthy population. We have not considered larger variance in the patient selection process but that is supported in the BioGears engine. Future work investigating how the patient population impacts patient trajectories is an interesting area of analysis and one that will be pursued by this team. Other impacts on trauma resuscitation, including how to consider comorbidities of unhealthy patients, is also omitted for this study but would be an excellent area of future work. Arterial pressure is used as a stabilization criterion by the engine before runtime. Convergence is performed by an augmented newton method using systemic resistance pathways as the iterative independent variables.

By considering a collection of maximum curvatures computed by a diverse patient set, we can compute a convex hull that provides the minimum geometric coverage, using the maximum curvature points as the object's vertices. We compute this shape using the scipy convex hull algorithm [61] to extract this shape from the data, Fig 6. The diverse patient data provides distinct shapes with moderate geometric coverage that provides a means towards quantifying patient data due to location in the state-space for a given trajectory. Top row denotes a moderate initial hemorrhage before resuscitation (100mL/min) with the bottom row being random patients initialized with a severe hemorrhage before resuscitation. We notice distinct differences between the area of coverage for the convex hull, as well as differences in maximum values. Overall physiology is more severe for a more severe hemorrhage with nearly no overlap between the two sets of bleeding patients. Arterial pH and lactate levels are extremely distinct with minimum pH achieved varying from 7.2 for the moderate patient set to 6.6 for the severe patient set. The corner of coverage for respiration rate has some overlap between the patient groups and more work is needed to understand distinct hull regions between patient groups. Future work will focus on simulating large populations over more refined hemorrhages to analyze statistical significance of the patient regions as a function of respiration rate and heart rate (two very prominent physiological markers for clinicians during a hemorrhagic event).

We note that generally the overall area covered by the computed hull is increased for the lower hemorrhage patients (100 mL/min on the top row), indicating less variability for a class 3 hemorrhage (200 mL/min bottom row). This variability seems qualitatively accurate given that as the physiology is stressed, patient physiology trends towards end-of-life characteristics, with less variation. By combining geometric space with the nonlinearity of the curvature, we can predict the hemorrhagic severity of the patient by using this coverage of the state-space. Future work may use this coverage as a marker for machine learning algorithms, as shape and region identification are a much more stable problem to solve than time-series estimation.

### 3.3 Validation of convex hull

We validate the regions of the convex hull by comparing experimental lactate values of the hemorrhage patient to the simulated regions of interest, Fig 7. For this analysis we create the convex hull that spans over lactate values as a function of

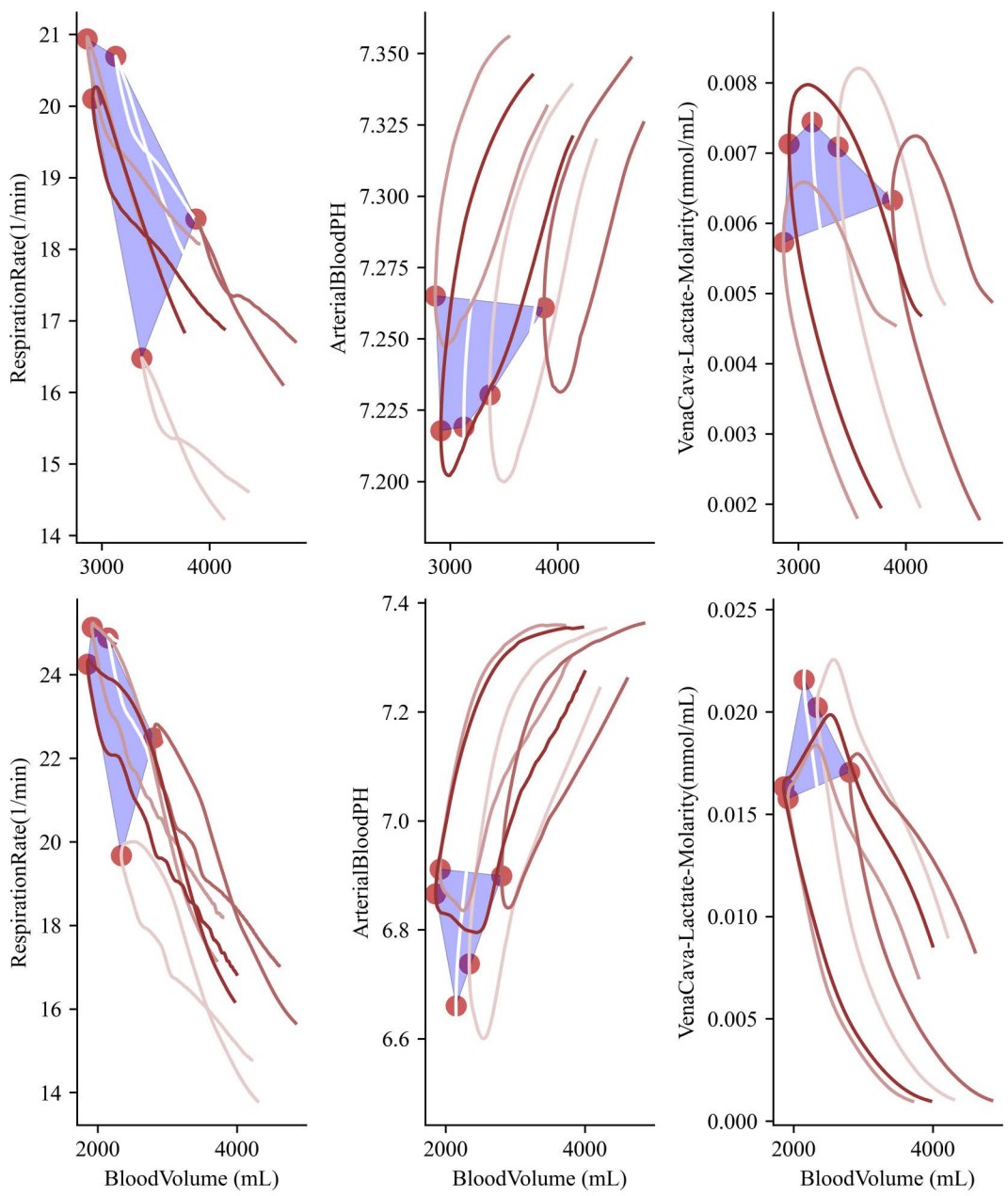

**Fig 6. Two sets of trajectories with the minimum convex hull computed with vertices at each maximum curvature point.** The top set are simulations initialized with 100 ml/min bleed rate and the bottom are patients initialized with a 200 ml/min bleed rate. Each trajectory corresponds to a distinct patient.

blood volume for 100, 150, and 200 mL/min initial hemorrhage rates. We then simulate the patient physiology for 5 random patients and collect the max curvature values over the simulation. For each group of 5 we compute the hull and plot. We note that this coverage is distinct for each patient cohort and subsequent initial hemorrhage rate. We collect experimental data for noted severity to compare simulated and real lactate values [6,15,62]. Simulated lactate values cover the variability and mean values of lactate reported in literature. We note that getting a specific blood volume for experimental

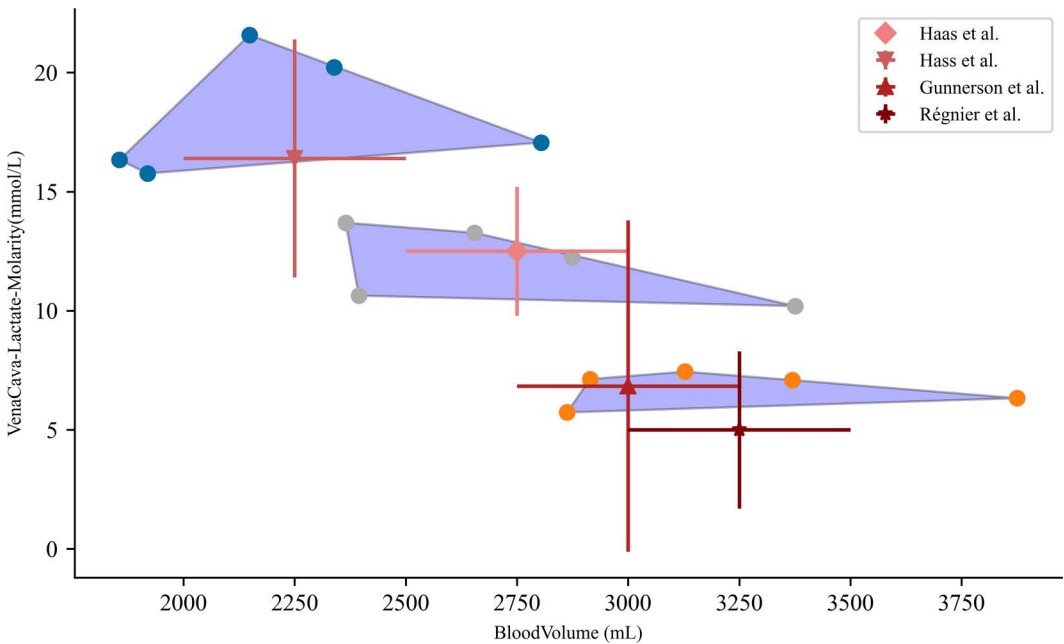

**Fig 7. Convex hulls computed for 5 patients with different bleed rates.** The left most hull corresponds to 200 mL/min, the middle to 150 mL/min, and the left most hull to 100 mL/min. Specific blood volume amounts were not reported in the experimental data, only "severity" of injury. We approximate severity with total blood volume for class 3, 2, and 1 hemorrhage.

data is not possible and so we estimate the blood volume by denoting severity of the trauma patient being reported in the data. Experimental data reports moderate, and mild trauma that we extrapolate to distinct blood volume values. These are just estimates but because the x-domain coverage of the hull spans a large distinct region we feel that this approximation is valid in this context. In addition, we create an x-domain span for this data of 500 mL for each severity classification and plot them with our simulated data. We note that the experimental data falls within the convex hull for a given lactate level and blood volume decrease. This is of note as the parameterization of the model was not directly fit to this experimental data, merely constructed from mechanisms of action and mathematical representations of these interactions. Ultimately, more validation must be performed on this model and will be an area of future work.

### 3.4 Curve fit to simulated data

Finally, we take our collection of 100 total simulated patients with varying levels of hemorrhage and extract their maximum curvature value and fit a logistic curve to classify the data, Fig 8. We see the nonlinearity of the patient physiology present in the curve fit, with the following regression curve fit to the data:

$$f(BV) = \frac{-19.0}{1 + exp\left(-0.36\left(BV - 9.38\right)\right)} + 24.5$$

Here BV denotes the maximum blood volume lost by the patient during the simulation. We may interpret this as a prediction of the patient lactate concentration given a maximal blood volume lost due to a hemorrhage. Clinically, we may also desire to use the inverse of this equation to approximate total blood volume lost in the patient given the measured serum lactate concentrations. This value may allow a clinician to derive a resuscitation scheme given these measurements. The statistical measurements of the fit and distribution of parameterization are as follows: std err of the

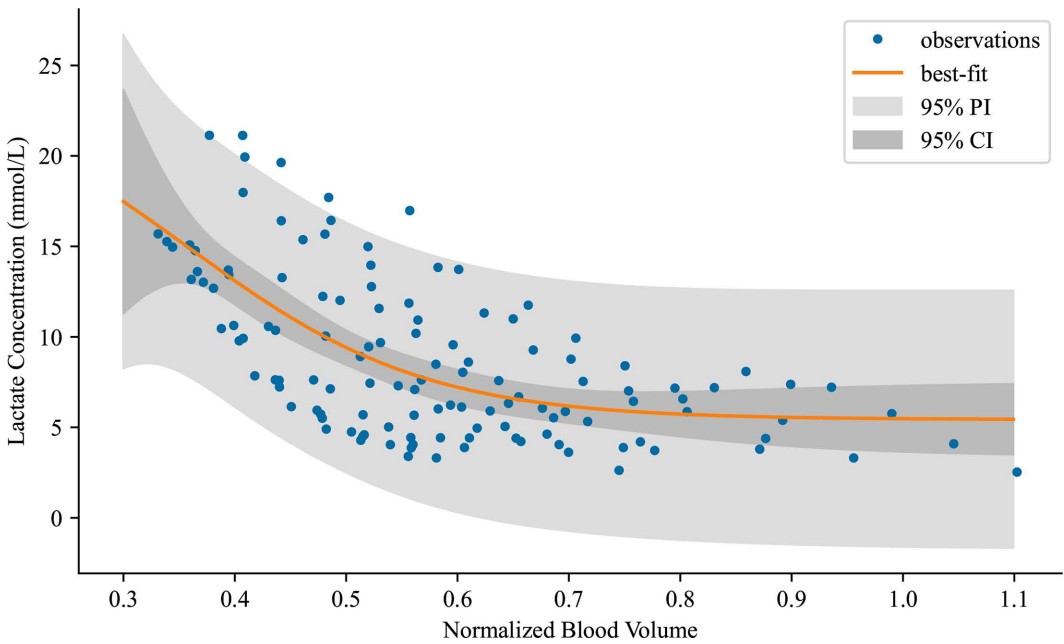

**Fig 8. Nonlinear logistic function fit to a collection of maximum curvature points over the entire simulated dataset.** We note a good fit up to the presented confidence interval.

regression = 3.4509, and the p-value of the regression F test = 7.25e-14. Statistical parameters paint a picture of a well fit curve that includes variance in the data. We note that nearly the entirety of the simulated data falls into our confidence interval of fit.

There are some discrepancies that appear in this curve fit. First, the variability of the patient data is large and contributes to a lower r squared value than we would expect. We note that this value is computed for the initial curve fit and does not include estimation with the included confidence interval or prediction interval. We note, that in Fig 8 the prediction interval almost entirely encapsulates the simulated data, leading to a very strong estimator given the uncertainty in the data. Only 5 data points are outside the computed prediction interval, computed from the square of the covariance matrix of the parameter estimates, leading to a 95% encapsulation of the data. Variability is something we not only expect in our simulated patients but is inherent in human physiology. We expect broad prediction intervals for any estimator given this variability. In addition, the logistic function being fit to the lactate levels of the simulated patients is not a representation of the complexities of lactate production during acute hemorrhage. We try and consider these complexities in the computational model that we used to generate this data. This function is merely the first step towards a decision support tool that could be used in a clinical setting, but we do not suppose that it captures all the complexities of the patient until the computational model is further expanded upon.

## 4 Discussions

Presented here is a model of cardiovascular circulation and subsequent hemorrhage, with associated tissue oxygenation and anerobic response to hypoperfusion. We show that the model captures major physiological markers of hemorrhage that are used clinically to determine patient state. We use the simulated model to compute trajectories of patient recovery and using smoothing and differential geometry identify regions of maximum curvature in these trajectories. We note that the point of maximum curvature is a good indicator of patient health during hemorrhage and subsequent resuscitation. We use these points to capture a convex hull for different physiology and use this region in state-space to validate our model

using experimental data. We note that these regions may be used going forward to classify patient state based upon clinical measurements. From the total collection of these curvature points we can fit a logistic curve to the data as a function of blood volume. Including the uncertainty in the parameterization we can capture trends for the simulated data, providing a decision support model that is easily interpretable.

We also acknowledge that several important physiological responses, such as renal function, angiotensin activity, and capillary refill, are not included in the model. The kidney's ability to buffer blood pH, effectively controlling the blood-gas binding curve, alone is worth investigating further and will be an area of future work. We note, that the BioGears engine is an open-source platform and encourage other researchers to continue its development in the framework of these limitations. Refinement of the cardiovascular resolution and how impacts of capillary refill, localized tissue hypoxia and region-specific lactate generation will be an area of future research.

In general, there are major areas of investigation in future work, using this model. Firstly, we sensitivity of the model needs to be better understood, not only in the system level modeling context but also within the patient cohort construction. The patient cohort statistics are arbitrarily narrow (to only include analysis of a distinct healthy population) but understanding how patient trajectories are impacted by various cohort distributions is an area of future work. In addition, although hypovolemic trauma in the ICU is often affecting healthy male adults, comorbidities play a pronounced role in hemorrhagic shock outcomes, especially in the context of surgy. Investigating outcomes of various comorbidities on the resuscitation of the patient is something we will investigate in future work. Parameterization and sensitivity of the model.

Ultimately, the logistic model, as a determinant of patient health, is only as good as the complexity of the models being used to produce simulated data. Currently, this model is missing some of that complexity but is merely a first step towards a model that may encapsulate the multi-system multi-scale response of the body to hemorrhagic trauma. We do not recommend this curve to be used as a decision support tool under these assumptions but do demonstrate some applicability in matching the validation data that we present here. We aim to continue to develop the BioGears platform to encapsulate the human physiological response to acute trauma in such a way that it may be a good tool as an in-silico patient surrogate for more sophisticated machine learning and deep learning applications. We are inspired by MuJoCo [63], which has been used as a biomechanics surrogate for robotic simulated training, as a template for the BioGears engine.

### 4.1 Lactate model for training and simulation

Physiology engines, such as BioGears, can play a fundamental role in creating accurate and realistic medical simulations, where changes in the patient's condition – be it either improvement or deterioration – rely on appropriate and timely interventions by the students. But the ability of an engine to represent physiological changes over time, as the "patient" moves through consecutive echelons of care, becomes especially useful for a specific area of medical simulation: training our militaries' combat medical personnel.

Medical interventions at the point of injury have become well established throughout the military with the implementation of Tactical Combat Casualty Care (TCCC) [64], focused on rapid hemorrhage control (via application of tourniquets or wound packing), airway support and breathing (via positioning the patient or performing emergency needle decompression), and the transfer of the patient to a higher level of care as soon as feasibly possible. In an early application of BioGears physiology to TCCC, Sims and Wentz used an earlier BioGears model to simulate vital signs and blood gas labs for multi-trauma female patients being treated at a Role 1 Field Hospital. During the 30 minutes that the patient was treated in the emergency room, heart rate, respiratory rate, and blood pressure changed appropriately in response to fluid transfusions, administration of oxygen, needle decompression of the chest, and other interventions. However, laboratory values did not exhibit the increasing lactate and decreasing pH that would be expected, and students did not perform the additional interventions that would have been needed to achieve desired outcomes. The modeling of lactate described in this paper improved the simulated response, such that impending acidosis was properly treated.

The battlefield is changing, and so too is the approach to treating the wounded. And as the military anticipates delays in patient transport to higher echelons of care, the means of training their medical personnel must also evolve to reflect this new treatment paradigm. Thus, the focus on Prolonged Casualty Care (PCC). The feasibility of the "Golden Hour," where trauma patients receive definitive treatment at a far forward medical facility within an hour of being injured, will be impossible, and combat medical personnel will be forced to provide care for extended periods of time (hours to even days). With PCC as the new normal in battlefield medicine, training and simulation must prepare providers for caring for seriously wounded patients over extended periods of time, using limited resources. Physiology engines that can simulate the dynamic physiological response to traumatic injury, such as fluctuating lactate levels, will help prepare personnel for these challenging situations.

Simulation training for PCC scenarios is becoming well established among military medical personnel.59 But one of the primary challenges of training for prolonged casualty care is what Pike and Mazzeo refer to as the "tyranny of time,"60 AKA, "How to compress a 48–72-hour PCC scenario into a 1–2-hour training period while capturing all the physiology changes…" This is where accurate physiology engines play a role. By fast-forwarding a patient state, students can assess and treat the simulated patient based on changes in physiological responses over time. Serial point of care testing of lactate levels has shown promise in predicting criticality of sepsis patients in emergency room settings,61 and can also be deployed in austere environments for use in trauma patients. Integrating this procedure in PCC simulations, supported by a physiology engine (such as BioGears) that can accurately represent fluctuations in lactate levels as the "patient" improves or deteriorates, can enhance training and prepare military medical personnel for the challenges that lay ahead.

### 4.2 Lactate as an indicator of patient state in the ICU

Lactate is a well-studied marker for tissue hypoxemia in shock settings for ICU patients with a variety of physiologic states. In hemorrhagic shock and trauma patients, elevated lactate has been shown to correlate with an increase in mortality. Specifically, the trend of lactate clearance has been suggested as a better marker of resuscitation success [65]. Patients in profound hemorrhagic shock with vital sign derangements will get blood product resuscitation, however, in patients with vital signs that have normalized, lactate can be utilized as a more sensitive marker of the success of the resuscitation. Oftentimes, patients will have normalization of vital signs prior to clearance of lactate once hemorrhage is controlled. In these patients, trending serial lactate levels can indicate whether tissue hypoperfusion is still ongoing, and thus they would benefit from additional transfusions [66]. This should help to mitigate this occult perfusion and thus decrease the risk of under-resuscitation for hemorrhagic shock after hemorrhage control is achieved. Lactate-guided resuscitation has been shown to decrease in-hospital mortality for multiple groups of patients and is a vital marker for appropriate resuscitation of hemorrhagic shock after trauma [67]. As renal clearance of lactate is robust, the trajectory of lactate as a marker for patient state and trajectory is well supported [6,7,13,55,68]. Although, there are other, additional, physiological details that may indicate the state of the patient that are not considered for this manuscript and would be beneficial to investigate such as: capillary refill time, cardiac output, neurological status, and comorbidities. Each of these warrants additional model development and analysis. In this sense a computational model of lactate production due to hypoperfusion is an excellent tool to discover latent states in the patient recovery and provide quantifiable direction pertaining to testing after bleeding has been controlled.

## 5 Conclusions

Here we present a whole-body physiology model of hemorrhage and subsequent hypovolemia blood substance markers. We show this model qualitatively fits physiology data of hemorrhage and use the output data to present an analysis of state-space as a decision support tool. This space validates with experimental data and can be used to interpret patient state going forward. We note that there are numerous other physiological responses not presented here such as renal, angiotensin, and capillary refill that are not modeled and may be areas of future work. We also note that state-space

curvature may be used in future machine learning classification models to determine patient hemorrhage severity. Classification tasks are much more robust than time-series prediction algorithms and so this approach, using simulated data, may be used as a data stream into such models. Future work may try and analyze performance between these algorithms.

## Acknowledgments

We'd like to acknowledge the support and guidance of Hugh Connacher, Harvey Magee, Dr. Brett Talbot, and Geoff Miller who all provided excellent guidance during the original development of the BioGears project. We'd also like to acknowledge critical members of the BioGears development team over the years: Matthew McDaniel, Nathan Tatum, Jenn Carter, Jeff Webb, Aaron Bray, and Rachel Clipp. We also thank Applied Research Associates for supporting this open-source platform and continuing to usher its software development and architecture to aid its use in physiology modeling and research.

We'd also like to acknowledge the support and guidance of Conner Parsey and Teresita Sotomayor of STTC, who identified the need for improved models of the effect of hypovolemia and emergency interventions on patient outcomes, with an emphasis on hyperlactatemia and acidosis.

## Author contributions

**Conceptualization:** Austin Baird, Erika K Bisgaard, Rachel K Wentz, Edward M Sims, David Hananel.

**Data curation:** Austin Baird, Steven A. White, David Hananel.

**Formal analysis:** Austin Baird, Erika K Bisgaard.

**Funding acquisition:** Rachel K Wentz, Edward M Sims, David Hananel.

**Investigation:** Austin Baird, Erika K Bisgaard, Rachel K Wentz.

**Methodology:** Austin Baird.

**Resources:** Austin Baird, Rachel K Wentz, Edward M Sims.

**Software:** Austin Baird, Steven A. White, Rachel K Wentz.

**Supervision:** Austin Baird, Edward M Sims, David Hananel.

**Validation:** Austin Baird, Steven A. White.

**Visualization:** Austin Baird.

**Writing – original draft:** Austin Baird, Erika K Bisgaard.

**Writing – review & editing:** Austin Baird, Steven A. White, Erika K Bisgaard, Rachel K Wentz, Edward M Sims, David Hananel.

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
