## [Decision Letter · Decision Letter 0]

1 May 2025

PCOMPBIOL-D-25-00012

A Model of Anaerobic Tissue Perfusion During Trauma –Lactate Trajectory Curvature Can Determine Recovery

PLOS Computational Biology

Dear Dr. Baird,

Thank you for submitting your manuscript to PLOS Computational Biology. After careful consideration, we feel that it has merit but does not fully meet PLOS Computational Biology's publication criteria as it currently stands. Therefore, we invite you to submit a revised version of the manuscript that addresses the points raised during the review process.

Please submit your revised manuscript within 60 days Jul 01 2025 11:59PM. If you will need more time than this to complete your revisions, please reply to this message or contact the journal office at ploscompbiol@plos.org. Please include the following items when submitting your revised manuscript:

We look forward to receiving your revised manuscript.

Kind regards,

Kevin Flores

Guest Editor

PLOS Computational Biology

Mark Alber

Section Editor

PLOS Computational Biology

**Journal Requirements:**

At this stage, the following Authors/Authors require contributions: Austin Baird, Steven A. White, Erika K Bisgaard, Rachel K Wentz, Edward M Sims, and David Hananel. Please ensure that the full contributions of each author are acknowledged in the "Add/Edit/Remove Authors" section of our submission form.

3) Some material included in your submission may be copyrighted. According to PLOSu2019s copyright policy, authors who use figures or other material (e.g., graphics, clipart, maps) from another author or copyright holder must demonstrate or obtain permission to publish this material under the Creative Commons Attribution 4.0 International (CC BY 4.0) License used by PLOS journals. Please closely review the details of PLOSu2019s copyright requirements here: PLOS Licenses and Copyright. If you need to request permissions from a copyright holder, you may use PLOS's Copyright Content Permission form.

Potential Copyright Issues:

- Figure 1. Please confirm whether you drew the images / clip-art within the figure panels by hand. If you did not draw the images, please provide (a) a link to the source of the images or icons and their license / terms of use; or (b) written permission from the copyright holder to publish the images or icons under our CC BY 4.0 license. Alternatively, you may replace the images with open source alternatives. See these open source resources you may use to replace images / clip-art:

4) Please amend your detailed Financial Disclosure statement. This is published with the article. It must therefore be completed in full sentences and contain the exact wording you wish to be published. Please ensure that the funders and grant numbers match between the Financial Disclosure field and the Funding Information tab in your submission form. Note that the funders must be provided in the same order in both places as well.

**Reviewers' comments:**

Reviewer's Responses to Questions

**Comments to the Authors:**

Reviewer #1: Overall the article was well constructed and the research was original. There were minor questions/concerns I had. Is this computational model done in 1D or is it a multi-domain model? I did not see this explicitly defined in the article and it would be helpful for readers to have this stated up front early on in the article. Below are minor edits, mainly concerning the text.

Line 177 - What is meant by "pressure head"? Do you mean cerebral pressure? There is also mention that this blood pressure does not affect the baroreceptor reflex when laying down, so are you taking gravity into consideration with patient-specific heights/vessel lengths and/or angles?

Line 179 - misspelled "use".

Line 224 - You say parameter values were chose to fit data well. This is very vague, how did you know the range of parameters to explore when determining these values? From literature? From previous studies? Was there any sensitivity analysis or parameter estimation done to determine these values?

Line 242 - Misspelled "etc."

Line 247 - "smoothest" seems to be incorrectly used.

Line 253 - "begins their recover" seems incomplete. Perhaps authors meant to say "begins to recover" or "begins their recovery".

Line 283 - "addition" should be "additional" in this context.

Line 458 - "emergence" should be "emergency" in this context.

Reviewer #2: The paper by Baird et al. presents a compelling study on anaerobic tissue perfusion during trauma. It introduces and analyzes a mathematical model of hemorrhagic shock and trauma, aiming to construct patient-specific trajectories of serum lactate. The ultimate goal is to support healthcare professionals in making more informed transfusion decisions.

The article addresses a highly relevant topic and presents promising results with significant potential for clinical application. The reviewer believes the paper can be accepted after the following minor revisions are addressed:

- The model includes several nonlinear feedback loops, but some of these are not easy to follow in the current presentation of Section 2. Including a block diagram or more descriptive figures would help clarify the interrelationships between variables across different components of the model.

- The authors utilize the BioGears engine; however, it is not entirely clear which equations are novel contributions and which are adopted from existing work. The authors should explicitly distinguish between pre-existing elements of BioGears and the modifications or additions made in this study. A clearer discussion of the model’s novel aspects is necessary.

- The selection of model parameters—whether explicitly stated or implied through numerical values in various equations—lacks sufficient justification. A more thorough discussion of parameter choices, their influence on outcomes, and whether they can be adapted to patient-specific contexts is essential. This would strengthen the assessment of the model’s potential impact in real-world clinical settings.

Reviewer #3: The authors present a computational model simulating hemorrhagic trauma and recovery, proposing that the curvature of lactate-pH trajectories in phase-space can serve as an indicator of patient recovery. Built on the BioGears platform, the model includes multiple organ system interactions and introduces a logistic curve fit to summarize severity. The study addresses an important clinical modeling gap and introduces some creative computational ideas. However, the validation is largely qualitative, the model modifications are not clearly delineated, the theoretical basis for the curvature approach is underdeveloped, and several methodological details are missing. The claims of clinical applicability are premature given the current evidence, and major revisions are needed to improve the rigor, clarity, and robustness of the work.

Strengths of the Manuscript

• The clinical relevance of modeling hemorrhagic shock and recovery is clear and important.

• The use of curvature of physiological trajectories is a novel computational approach in this setting.

• Leveraging and extending an open-source platform like BioGears promotes transparency and reproducibility.

• The authors are appropriately cautious in acknowledging limitations and outlining future work.

• The logistic curve fitting method provides a potentially useful way to summarize complex simulation results.

Major Criticisms and Questions for Authors

• The validation is qualitative and limited to endpoint lactate levels; can you provide quantitative metrics comparing simulated and experimental time-resolved lactate and pH data?

• The description of model extensions and modifications to BioGears is vague; can you provide a clear breakdown of what was added or modified compared to the original engine?

• As far as I can tell, the biological justification for using curvature to define recovery phases is missing; can you explain why maximum curvature points should correspond to transitions in patient state?

• The variability of patient physiology in simulations was artificially narrow; how would your results change with a broader range of patient characteristics such as comorbidities?

• The smoothing and curvature calculation methods may affect the results substantially; have you performed sensitivity analyses to different smoothing window sizes and curvature algorithms?

• Details of the simulated patient cohort, including the number of patients per hemorrhage severity and the use of random seeds, are unclear; can you provide more information on cohort design and statistical robustness?

• The literature review omits recent computational models of lactate kinetics and trauma physiology; incorporating studies such as Stefanovski et al. (2022, American Journal of Physiology - Endocrinology and Metabolism), Villota-Narvaez et al. (2023, Scientific Reports), and Brooks (2020, The Journal of Physiology) would strengthen the background and show how your work builds upon or differs from existing computational models.

**Have the authors made all data and (if applicable) computational code underlying the findings in their manuscript fully available?**

Reviewer #1: Yes

Reviewer #2: **No: ** The code does not seem to be publicly available.

Reviewer #3: Yes

PLOS authors have the option to publish the peer review history of their article (what does this mean? ). If published, this will include your full peer review and any attached files.

**Do you want your identity to be public for this peer review?** For information about this choice, including consent withdrawal, please see our Privacy Policy .

Reviewer #1: No

Reviewer #2: No

Reviewer #3: No

**Figure resubmission:**

**Reproducibility:**



---

## [Decision Letter · Decision Letter 1]

27 Jul 2025

Dear Dr. Baird,

We are pleased to inform you that your manuscript 'A Model of Anaerobic Tissue Perfusion During Trauma –Lactate Trajectory Curvature Can Determine Recovery' has been provisionally accepted for publication in PLOS Computational Biology.

Best regards,

Kevin Flores

Guest Editor

PLOS Computational Biology

Mark Alber

Section Editor

PLOS Computational Biology

Reviewer's Responses to Questions

**Comments to the Authors:**

Reviewer #1: The changes made have greatly strengthened the paper. I believe it should be accepted.

Reviewer #2: Authors answered successfully to all my previous remarks

Reviewer #3: The authors have addressed my previous comments.

**Have the authors made all data and (if applicable) computational code underlying the findings in their manuscript fully available?**

Reviewer #1: Yes

Reviewer #2: Yes

Reviewer #3: None

PLOS authors have the option to publish the peer review history of their article (what does this mean? ). If published, this will include your full peer review and any attached files.

**Do you want your identity to be public for this peer review?** For information about this choice, including consent withdrawal, please see our Privacy Policy .

Reviewer #1: No

Reviewer #2: No

Reviewer #3: No

---

## [Editor Report · Acceptance letter]

PCOMPBIOL-D-25-00012R1

A Model of Anaerobic Tissue Perfusion During Trauma –Lactate Trajectory Curvature Can Determine Recovery

Dear Dr Baird,

I am pleased to inform you that your manuscript has been formally accepted for publication in PLOS Computational Biology. Your manuscript is now with our production department and you will be notified of the publication date in due course.

With kind regards,

Anita Estes
